

# Systematic evaluation and meta-analysis of the prognosis of down-staging human papillomavirus (HPV) positive oropharyngeal squamous cell carcinoma using cetuximab combined with radiotherapy instead of cisplatin combined with radiotherapy

Qiong Hu, Feng Li and Kai Yang

Department of Oral and Maxillofacial Surgery, The First Affiliated Hospital of Chongqing Medical University, Chongqing, China

Corresponding author
Kai Yang, cqfyyk@hotmail.com

## ABSTRACT

**Objective**. To evaluate the efficacy and safety of cetuximab instead of cisplatin in combination with downstaging radiotherapy for papillomavirus (HPV) positive oropharyngeal squamous cell carcinoma (HPV$^+$ OPSCC).

**Design**. Meta-analysis and systematic evaluation.

**Data sources**. The PubMed, Embase, Web of Science, and Cochrane library databases were searched up to June 8, 2023, as well as Clinicaltrials.gov Clinical Trials Registry, China Knowledge Network, Wanfang Data Knowledge Service Platform, and Wipro-journal.com.

**Eligibility criteria for selecting studies**. Randomized controlled trials reporting results of standard regimens of cetuximab + radiotherapy vs cisplatin + radiotherapy in treating HPV$^+$ OPSCC were included. The primary outcomes of interest were overall survival (OS), progression-free survival (PFS), local regional failure rate (LRF), distant metastasis rate (DM), and adverse events (AE).

**Data extraction and synthesis**. Two reviewers independently extracted data and assessed the risk of bias of the included studies. The HR and its 95% CI were used as the effect analysis statistic for survival analysis, while the OR and its 95% CI were used as the effect analysis statistic for dichotomous variables. These statistics were extracted by the reviewers and aggregated using a fixed-effects model to synthesise the data.

**Results**. A total of 874 relevant papers were obtained from the initial search, and five papers that met the inclusion criteria were included; a total of 1,617 patients with HPV$^+$ OPSCC were enrolled in these studies. Meta-analysis showed that OS and PFS were significantly shorter in the cetuximab + radiotherapy group of patients with HPV$^+$ OPSCC compared with those in the conventional cisplatin + radiotherapy group (HR = 2.10, 95% CI [1.39–3.15], $P$ = 0.0004; HR = 1.79, 95% CI [1.40–2.29], $P$ < 0.0001); LRF and DM were significantly increased (HR = 2.22, 95% CI [1.58–3.11], $P$ < 0.0001; HR = 1.66, 95% CI [1.07–2.58], $P$ = 0.02), but there was no significant difference in overall grade 3 to 4, acute and late AE overall (OR = 0.86, 95% CI [0.65–1.13], $P$ = 0.28).

**Conclusions.** Cisplatin + radiotherapy remains the standard treatment for HPV$^+$ OPSCC. According to the 7th edition AJCC/UICC criteria, low-risk HPV$^+$ OPSCC patients with a smoking history of $\leq$ 10 packs/year and non-pharyngeal tumors not involved in lymphatic metastasis had similar survival outcomes with cetuximab/cisplatin + radiotherapy. However, further clinical trials are necessary to determine whether cetuximab + radiotherapy can replace cisplatin + radiotherapy for degraded treatment in individuals who meet the aforementioned characteristics, particularly those with platinum drug allergies.

**Prospero registration number.** CRD42023445619.

## BACKGROUND

The main risk factors for oropharyngeal squamous cell carcinoma (OPSCC) are smoking and alcohol abuse, which have strong synergistic effects (*Giraldi et al., 2017*; *Marziliano, Teckie & Diefenbach, 2020*). However, the incidence of OPSCC has been on the rise in recent years, and the incidence of OPSCC tends to be in young and middle-aged people, but most patients in Europe and the United States do not have these exposure factors. Epidemiologic and molecular biology studies have confirmed the association with HPV infection, especially high-risk HPV infection (*Muñoz et al., 2003*).

It has been shown (*Kelly et al., 2018*) that concomitant HPV infection status is an independent favorable prognostic factor, and that this type of cancer may be characterized by tumor biology with degradation of p53, inactivation of the Rb pathway, and up-regulation of p16, which improves susceptibility to radiation and chemotherapy. Patients with HPV$^+$ OPSCC who undergo a combination of conventional treatments such as surgery and radiotherapy generally have a better prognosis compared to patients with HPV- negative OPSCC (HPV$^-$ OPSCC) patients. Back in 2018, HPV$^+$ OPSCC was included as a separate entity in the 8th edition of the UICC/AJCC (International Union against Cancer/American Joint Committee on Cancer) staging manual (*NCCN, 2018*); not only was its TNM clinical staging downgraded, but it was also recommended that OPSCC be grouped in clinical practice according to the presence or absence of HPV infection.

Previously, in the clinic, regarding OPSCC, a combination of surgery, radiotherapy, and chemotherapy alone or in combination with targeted therapy was the mainstay of treatment (*Wang et al., 2020*; *Bourhis et al., 2006*). In December 2022, the National Comprehensive Cancer Network (NCCN) released the 1st edition of the NCCN Clinical Practice Guidelines for Head and Neck Cancer for 2023 (*Caudell et al., 2022*). In the new version of the guideline based on the AJCC (8th edition) TNM staging, some of the clinical staging of oropharyngeal cancer was changed, for example, the HPV$^+$ $T_0N_0$ stage was merged into the HPV$^+$ $T_1\sim_2N_0$ stage, and modified to the HPV$^+$ $T_0\sim_2N_0$ stage; and at the same time, the treatment strategy was further subdivided according to the size and number

of metastatic lymph nodes. Patients with $HPV^+$ $T_{0-2}N_1$ were treated with primary resection and ipsilateral or bilateral lymph node dissection, radical radiotherapy, concurrent systemic therapy/radiotherapy (level 2B evidence), or participation in clinical trials. If a patient had a single lymph node >3 cm or $\geq 2$ ipsilateral lymph nodes $\leq$ six cm, they were classified into the treatment strategy for patients with stage $T_{0-2}N_2$ and stage $T_3N_{0-2}$. The level of evidence for surgical methods remained unchanged. Concurrent systemic therapy and radiotherapy have been upgraded to Level 2A evidence. Radical radiotherapy has been replaced with post-induction chemotherapy (Level 3 evidence) or systemic therapy and radiotherapy. It is also clearly stated that, for OPSCC treatment (regardless of HPV infection), the new guidelines recommend the least treatment option to minimize the toxicity associated with treatment and to preserve function; triple therapy should be avoided as much as possible. For patients with head and neck cancer, the current clinical practice is mainly based on surgery and radiotherapy.

Due to the hidden location of OPSCC and numerous special anatomical structures, patients suffer not only from the serious complications brought about by open surgery but also from the serious adverse reactions brought about by postoperative combined with radiotherapy. After investigation, more than half of the patients, especially those with locally advanced disease, often choose non-surgical treatment (*Chang et al., 2017*), and through long-term observation and recording of radiation therapy, patients generally obtain good curative effects (*Golusiński & Golusińska-Kardach, 2019*). Compared with monotherapy, simultaneous treatment with chemotherapy and radiotherapy will have better clinical outcomes (*Lassen et al., 2022*; *Denis et al., 2004*), and currently, the most widely used cancer treatment is based on platinum-based drugs combined with radiotherapy, but the vast majority of patients experience serious adverse effects after treatment, including dry mouth syndrome, dysphagia, and hearing impairment. This treatment regimen, established before the prevalence of HPV-related cancers, may have over-treated this type of disease. *Mirghani & Blanchard (2018)* proposed the implementation of four downscaling treatment strategies for patients with $HPV^+$ OPSCC: (1) EGFR (Antiepidermal growth factor receptor) inhibitor combination radiotherapy instead of conventional cisplatin combination radiotherapy; (2) reduction of radiotherapy dose after induction chemotherapy; (3) emphasis on radiotherapy alone instead of simultaneous radiotherapy; (4) reduction of adjuvant radiotherapy dose after surgical treatment. Broadly speaking, $HPV^+$ OPSCC down-staging treatment strategies mainly include the drug substitution of cisplatin in combination with radiotherapy, reduction of radiotherapy dose or volume through combined treatment modalities to reduce/eliminate cytotoxic chemotherapy, and the use of non-invasive (minimally invasive) surgical procedures, among other avenues.

Comparatively, the latter three down-staging strategies are still mainly retrospective studies as well as single-arm trials, and there is a temporary lack of a sufficient amount of prospective randomized controlled trials for us to further analyze and study. In 2006, cetuximab, a chimeric monoclonal antibody against EGFR, was announced by the United States Food and Drug Administration (the FDA) to be approved for use in HNSCC (Head and Neck Squamous Cell Carcinoma) patients (*Cohen, 2014*). Cetuximab, as a monoclonal antibody that inhibits cancer cell growth factors, has greater selectivity for tumors, and

it kills cancer cells without affecting normal tissues too much, which can reduce the side effects and complications of treatment. A clinical trial (*Bonner et al., 2006*) showed that the use of cetuximab + radiotherapy regimen significantly improved the survival time of patients with HNSCC compared to conventional radiotherapy. The median survival of patients increased from the original 29.3 months to 49.0 months (5-year absolute survival rate of 9.2%) without increasing the common toxic effects of radiotherapy on the head and neck, and a similar benefit was shown in subgroup analysis for OPSCC (*Machtay et al., 2008*). Cetuximab + radiotherapy was superior to radiotherapy alone, improving local disease control (HR 0.68, $P = 0.005$), disease-free survival (HR 0.70, $P = 0.006$), and overall survival (HR 0.74, $P = 0.03$). Based on the results of this clinical trial by *Bonner et al. (2006)*, the cetuximab + radiotherapy regimen is likely to become the standard chemotherapy option for new first-line treatment of OPSCC. RTOG 1016, as the first randomized trial investigating the use of cetuximab + radiotherapy instead of cisplatin + radiotherapy for down-staging of patients with HPV$^+$ OPSCC (*Rischin et al., 2021*), has shown that that, although it could not reduce its toxic side effects, it significantly improved the survival of patients. According to the results of a subsequent subgroup analysis, after using cetuximab + radiotherapy instead of cisplatin + radiotherapy, HPV$^+$ OPSCC patients with the characteristics of age <65 years and tumor primary site located in the oropharynx could obtain better survival results. Meanwhile, in a meta-analysis report that included a total of 31 retrospective and some prospective studies (*Huang et al., 2016*), patients with HPV$^+$ OPSCC had a better prognosis in the cetuximab + radiotherapy group compared with cisplatin + radiotherapy. All this suggests that HPV$^+$ OPSCC is sensitive to cetuximab and the use of cetuximab + radiotherapy may become one of the options for downstaging this type of tumor.

Due to the youthful nature of HPV$^+$ OPSCC patients (*Psyrri et al., 2012*), survival is no longer just the only goal we pursue, but the quality of life of patients after treatment has become crucial. The rationale for downscaling therapy is precisely based on the high sensitivity of this type of cancer to radiotherapy, without compromising the standard treatment outcome in the patient population, to reduce the adverse effects caused by the treatment and thus improving the prognostic quality of life in HPV-related cancers. However, conclusions contrary to the results of the above clinical trials were obtained in several subsequent prospective randomized controlled trials, *i.e.*, the use of cetuximab instead of cisplatin combined with radiotherapy significantly reduces the survival of patients (*Buglione et al., 2017*; *Gillison et al., 2019*; *Mehanna et al., 2019*; *Gebre-Medhin et al., 2021*). Therefore, it is necessary to clarify the efficacy and safety of the cetuximab + radiotherapy regimen compared with the conventional cisplatin + radiotherapy regimen for the treatment of HPV$^+$ OPSCC and to perform a systematic evaluation and Meta-analysis.

In this study, we have conducted a systematic evaluation and Meta-analysis of the currently published studies related to cetuximab + radiotherapy regimens, using OS, PFS, LRF, DM, and AE as outcome indicators of effectiveness and safety, respectively, to analyze the effectiveness and safety of the cetuximab + radiotherapy regimen compared with the conventional cisplatin + radiotherapy regimen in the treatment of HPV$^+$ OPSCC, with the expectation of providing more effective and safe treatments for patients with HPV$^+$

OPSCC than for those with cisplatin. To analyze the efficacy and safety of the cetuximab + radiotherapy regimen compared with the traditional cisplatin + radiotherapy regimen in the treatment of HPV$^+$ OPSCC, it is expected to provide corresponding reference and guidance for the cetuximab + radiotherapy regimen in the treatment of HPV$^+$ OPSCC clinical practice.

## MATERIALS & METHODS

### Literature search strategy

This study was conducted in accordance with the Preferred Reporting Items for Systematic Reviews and Meta-Analyses (PRISMA) guidelines (*Page et al., 2021*). It has been registered with PROSPERO under the registration number CRD42023445619. A comprehensive computerized search was performed on PubMed, Embase, Cochrane Library, Web of Science databases, Clinicaltrials.gov, China Knowledge Network (CKN), Wanfang Data Knowledge Service platform, Wiprojournal.com using a combination of subject and free words: "Oropharyngeal Carcinoma", "Alphapapillomavirus", and "Controlled Trial". The search covered articles published between July 2, 2013 until June 8, 2023. Additionally, ongoing randomized controlled trials were reviewed to ensure inclusion of up-to-date results. Relevant references were also examined. Detailed search strategies are provided in Table 1 and Fig. 1 summarizes the risk of bias in the literature.

### Inclusion and exclusion criteria

Inclusion criteria: 1. Study subjects: (1) Patient greater than or equal to 18 years of age; (2) Patients' smoking history can be obtained; (3) Primary patients who have not been treated for related diseases before enrollment and who do not have distant metastases; (4) Bone marrow, liver, and renal function are good; (5) At least P16-positive permanent overexpression in the RNA scope E6/E7 mRNA needs to be immunohistochemistry and/or confirmed Expression to be diagnosed as HPV$^+$ OPSCC patients, regardless of their classification.HPV status needs to be determined by histopathology through the P16 expression of immunohistochemical markers. Tumors were classified as P16-positive and confirmed to be of HPV status if ≥70% of tumor cells had significant and diffuse nuclear as well as cytoplasmic staining (*Jordan et al., 2012*); 2. Type of study: randomized controlled trial (RCT); 3. Intervention: The expected total injection dose of cetuximab was 2150 mg/m$^2$ (specific course of treatment and single injection dose were not limited) + a total radiotherapy dose of 70Gy (35 times for 6 weeks, 6 times a week); 4. Control group setting: The expected total injection dose of cisplatin was 2150 mg/m$^2$ (specific course of treatment and single injection dose were not limited)+ a total radiotherapy dose of 70Gy (35 times for 6 weeks, 6 times a week); 5. Outcome metrics: at least one of the following outcome metrics was included, OS: the time from randomization grouping to death due to any overall survival; PFS: time from randomization to death from any cause; LRF: time from randomization to first progression; DM:from randomized grouping to the appearance of continued growth distant from other sites. Incidence of AE: the type and degree of event grading were categorized according to the 4th edition of the CTCAE (*Bennett et al., 2016*); acute toxicity was defined as an adverse event that first appeared during treatment or within

**Table 1  Literature search criteria.**

**Literature Search Criteria**: searches were conducted using a combination of subject terms and free words: 'oropharyngeal cancer', 'metaplasmoma virus', 'controlled trials', all with a library cut-off date of June 8, 2023. For example, PubMed covers articles published between July 2, 2013 and June 8, 2023.

**PUBMED:**

(((("oropharyngeal squamous cell carcinoma"[Title/Abstract] OR "head and neck squamous cell carcinomas"[Title/Abstract] OR "squamous cell carcinoma head and neck"[Title/Abstract] OR "squamous cell carcinoma of the head and neck"[Title/Abstract] OR "head and neck squamous cell carcinoma"[Title/Abstract] OR "HNSCC"[Title/Abstract] OR "carcinoma squamous cell of head and neck"[Title/Abstract] OR "squamous cell carcinoma of the larynx"[Title/Abstract]) AND "Alphapapillomavirus"[Title/Abstract]) OR "Alphapapillomaviruses"[Title/Abstract] OR "human papillomavirus"[Title/Abstract] OR "human papillomaviruses"[Title/Abstract] OR "papillomavirus human"[Title/Abstract] OR "papillomaviruses human"[Title/Abstract] OR "hpv human papillomavirus"[Title/Abstract] OR "hpv human papillomaviruses"[Title/Abstract]) AND ((y_10[Filter]) AND (clinicaltrial[Filter])))

**Web of science:**

| ID | Search |
|---|---|
| #1 | (((((TS=(oropharyngeal squamous cell carcinoma)) OR TS=(head and neck squamous cell carcinomas)) OR TS=(squamous cell carcinoma head and neck)) OR TS=(squamous cell carcinoma of the head and neck)) OR TS=(head and neck squamous cell carcinoma)) OR TS=(carcinoma squamous cell of head and neck)) OR TS=(squamous cell carcinoma of the larynx)) |
| #2 | (((((((TS=(Alphapapillomavirus)) OR TS=(Alphapapillomaviruses)) OR TS=(human papillomavirus)) OR TS=(human papillomaviruses)) OR TS=(papillomavirus human)) OR TS=(papillomaviruses human)) OR TS= (hpv human papillomavirus)) OR TS=(hpv human papillomaviruses) |
| #3 | (((TS=(Randomized Controlled Trial)) OR TS=(controlled clinical trial)) OR TS=(randomized)) OR TS=(randomised) |
| #4 | #1 AND #2 AND #3 |

('oropharyngeal squamous cell carcinoma'/exp OR 'head and neck squamous cell carcinomas':ti,ab OR 'squamous cell carcinoma head and neck':ti,ab OR 'squamous cell carcinoma of the head and neck':ti,ab) OR 'head and neck squamous cell carcinoma':ti,ab OR 'carcinoma squamous cell of head and neck':ti,ab OR 'squamous cell carcinoma of the larynx':ti,ab) AND 'Alphapapillomavirus':ti,ab AND ('Alphapapillomaviruses':ti,ab OR 'human papillomavirus':ti,ab OR 'human papillomaviruses':ti,ab OR 'papillomavirus human':ti,ab OR 'papillomaviruses human':ti,ab OR 'hpv human papillomavirus':ti,ab OR 'hpv human papillomaviruses':ti,ab) AND [clinical trial]/lim)

**Cochrane:**

| ID | Search |
|---|---|
| #1 | MeSH descriptor:[oropharyngeal squamous cell carcinoma]explode all trees |
| #2 | (Alphapapillomavirus):ti,ab,kw OR (HPV):ti,ab,kw |
| #4 | #1 AND #2 |

3 months after treatment; late toxicity was defined as an adverse event that first appeared if the toxicity persisted or occurred after 3 months up to 24 months after treatment. Multiple occurrences of a single toxicity type within the period analyzed were counted as a single event and were counted only once when analyzing the total number of acute and late adverse events.

   Exclusion criteria: (1) Included studies of patients without histopathologic diagnosis; (2) Recurrent patients who had been treated for related diseases before enrollment and those who had distant metastases; (3) Studies are published in various forms, including reviews, systematic reviews, meta-analyses, letters, and case reports; (4) In vitro and animal experimental studies; (5) Literature that had incomplete raw data without at least two years of postoperative followup information; (6) Inclusion of the fullest and newest if duplicate cases were involved in different articles to ensure that there was no overlap of the number of cases in duplicate literature.

## Data extraction and quality evaluation

Two researchers (Qiong Hu and Feng Li) conducted literature screening and data extraction respectively. They excluded literature that did not meet the inclusion criteria and obtained the full text of eligible articles, screened out the literature and extracted key information including first author, study type, disease stage, treatment regimen, gender, mean age, ECOG score, overall survival (OS), progression-free survival (PFS) as an outcome measure. Additionally, they collected data on incidence of PFS, locoregional failure (LRF), distant metastasis (DM), and adverse effects (AEs). The Cochrane risk bias assessment tool was used by both researchers to independently assess article bias. After cross-checking their assessments with each other's findings and resolving any conflicts with assistance from a third researcher (Kai Yang). The evaluation mainly focused on seven items: random sequence generation; allocation concealment; blinding of participants and personnel; blinding of outcome assessment; completeness of outcome data; selective reporting; and other sources of bias.

## Statistical analysis

The analyses were performed using Review Manager V.5.4 (Cochrane Collaboration). The hazard ratio (HR) and its corresponding 95% confidence interval (CI) were utilized as the effect analysis statistics for survival analysis, while the odds ratio (OR) and its corresponding 95% CI were employed as the effect analysis statistics for dichotomous variables. The heterogeneity of the included studies was assessed using the $\chi^2$ test, with $I^2$ used to quantify it: if $P > 0.1$ and $I^2 \leq 50\%$, low heterogeneity among studies was considered, leading to a fixed-effects model analysis; if $P \leq 0.1$ and $I^2 > 50\%$, high heterogeneity among studies was considered, resulting in a random-effects model analysis; sensitivity analysis was conducted for effect sizes exhibiting significant heterogeneity to evaluate study reliability. Egger's test was applied to assess publication bias in the literature. The significance level for meta-analysis testing was set at $P = 0.05$.

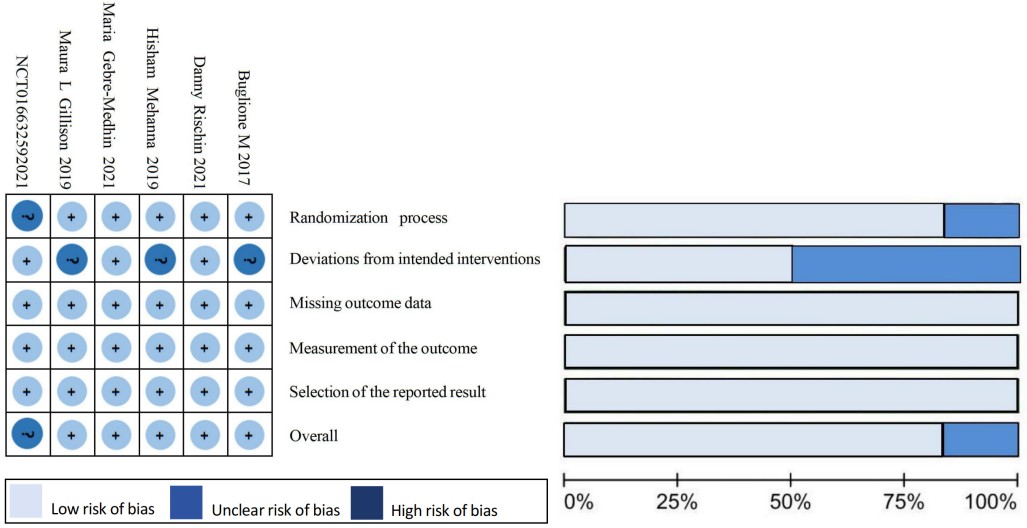

**Figure 1** Summary of results from the assessment of studies using the Cochrane Risk of Bias Tool.

# RESULTS

## Literature screening process and search results

After completing the initial search based on the specified search strategy, a total of 874 relevant literatures were obtained, eight duplicate literatures were removed, and the initial screening and full-text screening were completed according to the inclusion and exclusion criteria, and a total of five literatures that met the criteria were ultimately included (*Rischin et al., 2021*; *Buglione et al., 2017*; *Gillison et al., 2019*; *Mehanna et al., 2019*; *Gebre-Medhin et al., 2021*), including 1,617 cases of patients. The literature screening process and results are shown in Fig. 2.

## Basic characteristics of the included literature

The final five literature sources included in this study were all randomized controlled trials comparing cetuximab + radiotherapy to cisplatin + radiotherapy regimens for the treatment of HPV$^+$ OPSCC patients. Ongoing prospective trials without reported data and a single-arm trial without a control group (NCT01663259) were excluded. The literature reviewed in this study includes five clinical trials conducted between 2017 and April 2021. One of these trials was a phase II clinical trial (*Buglione et al., 2017*), while the remaining four were phase III clinical trials (*Rischin et al., 2021*; *Buglione et al., 2017*; *Gillison et al., 2019*; *Mehanna et al., 2019*; *Gebre-Medhin et al., 2021*). In total, the trials involved 1,617 subjects, with 806 receiving cetuximab + radiotherapy regimen and 811 receiving cisplatin + radiotherapy regimen. The Cochrane risk of bias assessment tool assessed the risk of bias in the literature as "low risk" and "some risk". This indicates that the quality of the included literature was high. Table 2 shows the basic characteristics of the included literature.
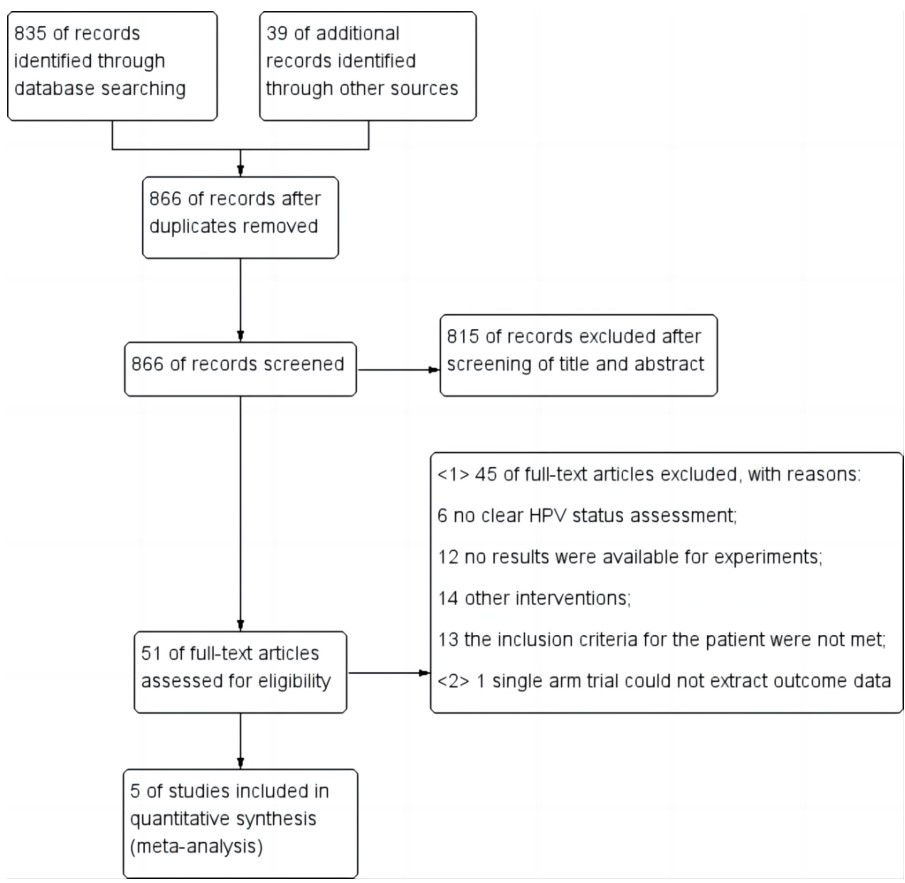

**Figure 2  Flow diagram for study selection.**

## Meta-analysis results
### OS
A total of five articles were included (*Rischin et al., 2021*; *Buglione et al., 2017*; *Gillison et al., 2019*; *Mehanna et al., 2019*; *Gebre-Medhin et al., 2021*), including 806 cases in the cetuximab + radiotherapy group and 811 cases in the cisplatin + radiotherapy group. A random-effects model was used to analyze the results, which showed that OS was significantly shorter in patients in the group using cetuximab + radiotherapy compared with the cisplatin + radiotherapy group (HR = 2.10, 95% CI [1.39–3.15], $P = 0.0004$). Further subgroup analysis based on the first intravenous input of cisplatin dose in the control group showed that OS was significantly prolonged in patients in the combination radiotherapy group using the first dose of cisplatin dose of 70 mg/m$^2$ (HR = 2.11, 95% CI [1.39–3.21], $P = 0.0005$), but the first dose of cisplatin dose of 100 mg/m$^2$ combined with radiotherapy group and cetuximab combined with radiotherapy group OS was not significantly different (HR = 2.41, 95% CI [0.73–7.96], $P = 0.15$), and its heterogeneity was obvious ($P = 0.03$, I$^2 = 78\%$) (Fig. 3).

Subgroup analysis was performed on age of patients (*Rischin et al., 2021*; *Gillison et al., 2019*; *Gebre-Medhin et al., 2021*) (635 cases in the cetuximab group: 643 cases in the

**Table 2  Characteristics of the included studies.**

| Reference source | Time limit | Experimental phase | Target group | Medicines | Dose/treatment | Ages | Total number of patients | Percentage of females |
|---|---|---|---|---|---|---|---|---|
| Buglione M | 2017 | II | T3-4, N0, M0 Any T, N+, M0 (except T1, N1) | cetuximab | Cetuximab 400 mg/m$^2$ loading dose 1 week prior to radiotherapy, followed by 250 mg/m$^2$ infusion during 7 weekly radiotherapy sessions | 62.5 | 9 | / |
| | | | | cisplatinum | During 7 weeks of RT, cisplatin was administered intravenously at a dose of 40 mg/m$^2$ with a maximum dose of 70 mg/m$^2$ | 70.5 | 9 | |
| Maura L Gillison | 2019 | III | Low risk is ≤ 10 pack-years (any N stage) or >10 pack-years and N0-N2a Moderate risk is >10 pack-years and N2b-N3 | cetuximab | Cetuximab 400 mg/m$^2$ loading dose 1 week prior to radiotherapy, followed by 250 mg/m$^2$ infusion during 7 weekly radiotherapy sessions | 57.4 | 399 | 10.00% |
| | | | | cisplatinum | Cisplatin 100 mg/m$^2$ on days 1 and 22 of radiotherapy (total 200 mg/m$^2$ ) | 57.7 | 406 | |
| Hisham Mehanna | 2019 | III | T1-T2 N2-N3 | Cetuximab cisplatinum | Intravenous cetuximab 400 mg/m$^2$ loading dose 1 week prior to radiotherapy, followed by 250 mg/m$^2$ infusion during 7 weekly radiotherapy sessions | 57 | 162 | 20.00% |
| | | | | Cetuximab cisplatinum | Three doses of intravenous cisplatin 100 mg/m$^2$ on days 1, 22, and 43 of radiotherapy, followed by 250 mg/m$^2$ during 7 weekly radiotherapy infusions | 56.5 | 159 | |
| Maria Gebre-Medhin | 2021 | III | T1-T2 T3-T4 | cetuximab cisplatinum | Cetuximab 400 mg/m$^2$ loading dose 1 week prior to radiotherapy, followed by 250 mg/m$^2$ infusion during 7 weekly radiotherapy sessions | 57 | 162 | 20.00% |
| | | | | Cetuximab cisplatinum | During 7 weeks of RT, cisplatin was administered intravenously at a dose of 40 mg/m$^2$ with a maximum dose of 70 mg/m$^2$ | 56.5 | 159 | |
| Danny Rischin | 2021 | III | Exclusion of T1-2N1 or stage IV (T4 and/or N3 and/or N2b-c if smoking history >10 pack-years and/or distant metastases) | cetuximab cisplatinum | During 7 weeks of RT, cisplatin was administered intravenously at a dose of 40 mg/m$^2$ with a maximum dose of 70 mg/m$^2$ | 56.5 | 90 | 10.00% |
| | | | | Cetuximab cisplatinum | During 7 weeks of RT, cisplatin was administered intravenously at a dose of 40 mg/m$^2$ with a maximum dose of 70 mg/m$^2$ | 58.1 | 92 | |

cisplatin group); ECOG score, smoking history, clinical stage of different types of tumors (*Rischin et al., 2021*; *Gillison et al., 2019*) (489:498) and primary tumor location (*Buglione et al., 2017*; *Gebre-Medhin et al., 2021*) (171:168). It has showed that age, T stage of the primary tumor, N stage of the primary tumor according to the 7th/8th edition of the

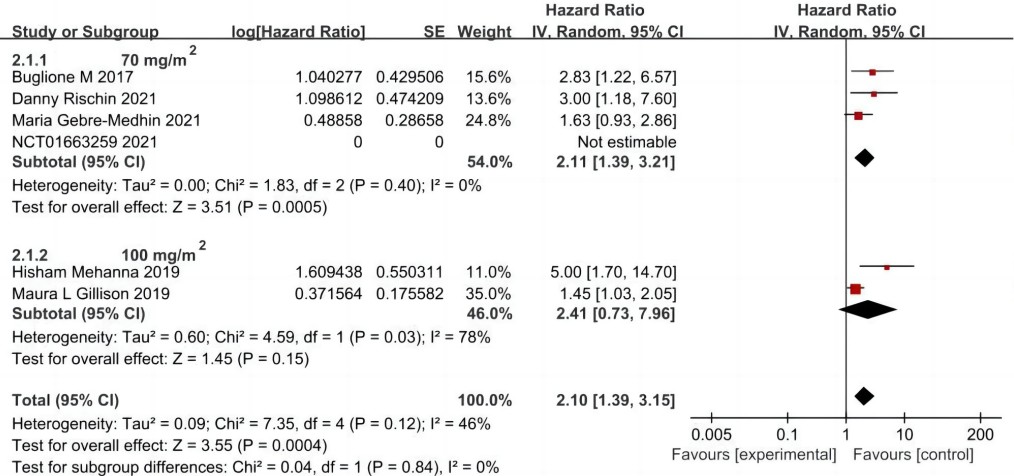

**Figure 3** Total survival forest map and quality assessment map.

UICC/AJCC definitions, and primary tumor stage according to the 8th edition of the UICC/AJCC definitions had a significant effect on OS. Using 65 years as the cut-off, the younger group of patients $\leq 65$ years (HR = 1.63, 95% CI [1.15–2.32], $P = 0.006$) compared to the group of patients > 65 years (HR = 1.48, 95% CI [1.06–2.05], $P = 0.02$) in terms of prolongation of patients' OS by cisplatin + radiotherapy regimen compared to cetuximab + radiotherapy regimen, had a more prominent advantage, *i.e.*, the younger patient population tended to have a higher survival rate, which is consistent with previous epidemiologic surveys and relevant retrospective reports.

In the $T_{1-2}$ stage low-risk group (*Ang et al., 2010*) (HR = 1.83, 95% CI [0.70–4.75], $P = 0.22$), the 7th edition $N_{2b-3}$ intermediate-risk group (HR = 1.94, 95% CI [0.78–4.86], $P = 0.16$), and the 8th edition staging group of patients with stage III (HR = 1.33, 95% CI [0.57–3.13], and $P = 0.51$), it has shown that treatment with cisplatin or cetuximab combined with radiotherapy had no significant effect on patients' OS. However, for the $T_{3-4}$ intermediate-risk group (HR = 1.64, 95% CI [1.34–2.02], $P < 0.00001$), the $N_{0-2a}$ stage low-risk in the 7th edition (HR = 1.88, 95% CI [0.7–4.75], $P = 0.002$), the $N_{0-3}$ in the eighth edition (HR = 1.46, 95% CI [1.25–1.71], and $P < 0.00001$), and in the group of stage I patients (HR = 1.59, 95% CI [1.18–2.14], $P = 0.002$) *versus* the group of stage II patients (HR = 1.41, 95% CI [1.10–1.81], $P = 0.006$) in the 8th edition of the staging, the use of cetuximab + radiotherapy compared cisplatin + radiotherapy significantly decreased the groups of OS (Figs. S1, S2, S3A).

Subgroup analyses based on ECOG score, smoking history, and primary tumor site showed no significant effect on patients' OS. In the ECOG = 1 subgroup (HR = 2.61, 95% CI [2.01–3.38], $P < 0.00001$), smoking history > 10 packs/year subgroup (HR = 1.62, 95% CI [1.28–2.05], $P < 0.00001$), and pharyngeal (HR = 2.17, 95% CI [1.12–4.22], $P = 0.02$), the cetuximab + radiotherapy regimen significantly reduced patients' OS compared with the cisplatin + radiotherapy regimen; in the ECOG = 0 subgroup (HR = 1.73, 95% CI [0.55–5.45], $P = 0.35$), smoking history $\leq 10$ packs/year subgroup (HR = 2.27, 95% CI

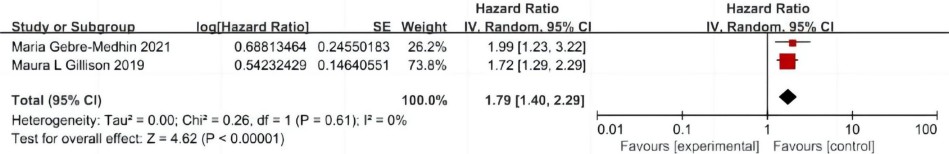

**Figure 4  Forest plot of progression-free survival.**

[0.80–6.42], $P = 0.12$), and the non-pharynx (HR = 1.09, 95% CI [0.54–2.21], $P = 0.81$) in which treatment with cisplatin or cetuximab did not have a significant differential effect on patients' OS (Figs. S3B, S4).

However, when the above factors were combined in individual subgroups, the results indicated that patients' OS was not associated with ECOG score, smoking history, or primary tumor site. Contrary to our previous perception, ECOG score and smoking history are not significant risk factors affecting OS. This may be due to the experimental group's attempt to circumvent the poor physical condition of ECOG = 1 patients themselves and smoking as a high-risk factor, which could not accurately reflect the impact of different therapeutic agents on survival outcomes. For the study, the low-risk group selected had a history of smoking ≤10 packs/year to minimize interference from other factors. However, this selection bias may have contributed to the observed heterogeneity.

## PFS

A total of two articles were included (*Gillison et al., 2019*; *Gebre-Medhin et al., 2021*), including 545 patients with cetuximab + radiotherapy and 551 patients with cisplatin + radiotherapy, and there was no significant difference in the test of heterogeneity between studies ($P = 0.61$, $I^2 = 0\%$), and a fixed-effect model was used. The analysis showed that patients in the cetuximab + radiotherapy group had significantly lower PFS compared with the cisplatin + radiotherapy group (HR = 1.79, 95% CI [1.40–2.29], $P < 0.0001$) (Fig. 4).

## LRF

A total of three articles were included (*Rischin et al., 2021*; *Gillison et al., 2019*; *Gebre-Medhin et al., 2021*), including 635 patients treated with cetuximab + radiotherapy and 643 patients treated with cisplatin + radiotherapy, and the test of heterogeneity between studies was not significantly different ($P = 0.76$, $I^2 = 0\%$), and a fixed-effects model was used. The analysis showed that the incidence of LRF was more than twice as high in patients in the cetuximab + radiotherapy group as in the cisplatin + radiotherapy group (HR = 2.22, 95% CI [1.58–3.11], $P < 0.0001$) (Fig. 5).

## DM

A total of three articles were included (*Rischin et al., 2021*; *Gillison et al., 2019*; *Gebre-Medhin et al., 2021*), including 635 patients in the cetuximab + radiotherapy group and 643 patients in the cisplatin + radiotherapy group, with no significant difference in the heterogeneity test between studies ($P = 0.32$, $I^2 = 12\%$), and a fixed-effect model was used. The analysis showed that DM was significantly higher in patients in the cetuximab

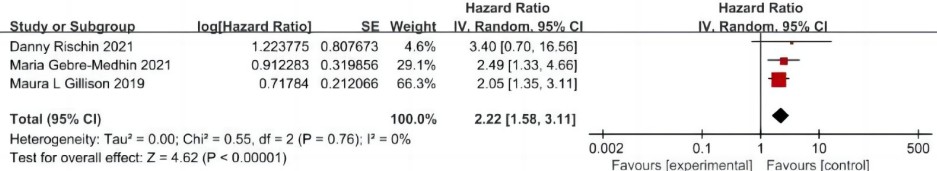

**Figure 5  Local area failure rate forest map.**

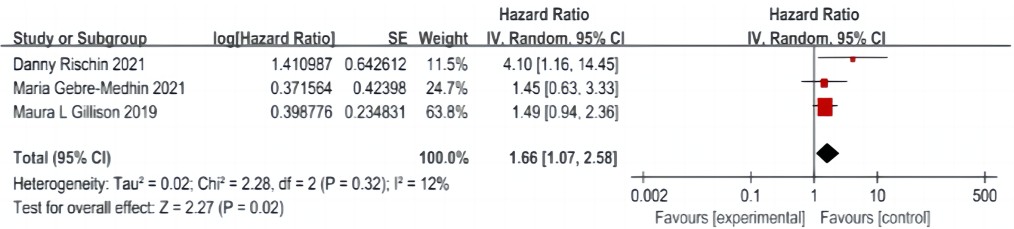

**Figure 6  Distant transfer rate forest map.**

+ radiotherapy group than in the cisplatin + radiotherapy group (HR = 1.66, 95% CI [1.07–2.58], $P = 0.02$) (Fig. 6).

### AE

A total of two papers were included (*Gillison et al., 2019*; *Mehanna et al., 2019*), including 561 patients in the cetuximab + radiotherapy group and 565 patients in the cisplatin + radiotherapy group, and the heterogeneity test was not significantly different between the studies ($P = 0.27$, $I^2 = 17\%$), with an $I^2$ of < 50% and a fixed-effect model was used. The analysis showed that the overall AE incidence in the cetuximab + radiotherapy group was not significantly different from that in the cisplatin + radiotherapy group (OR = 0.86, 95% CI [0.65–1.13], $P = 0.28$) (Fig. 7).

Incidence of grade III-IV AE: A total of two articles were included (*Gillison et al., 2019*; *Mehanna et al., 2019*), including 561 patients in the cetuximab + radiotherapy group and 565 patients in the cisplatin + radiotherapy group. (1) Acute incidence of grade III-IV AE: There was no significant difference in the test of heterogeneity between studies ($P = 0.79$, $I^2 = 0\%$), $I^2 < 50\%$, and a fixed-effects model was used. The analysis showed that the overall AE incidence rate in the cetuximab + radiotherapy group was not significantly different from that in the cisplatin + radiotherapy group (OR = 0.85, 95% CI [0.62–1.18], $P = 0.34$) (Fig. S5A).

(2) Incidence of late grade III-IV AE: no significant difference between studies by heterogeneity test ($P = 0.42$, $I^2 = 0\%$), $I^2 < 50\%$, using a fixed-effects model. The analysis showed that the overall AE incidence rate in the cetuximab + radiotherapy group was not

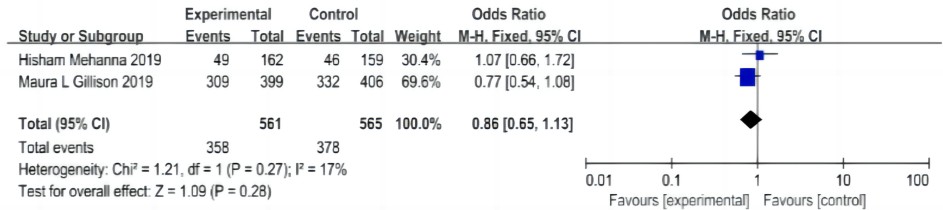

**Figure 7** Forest plot of adverse effects.

significantly different from that in the cisplatin + radiotherapy group (OR = 0.80, 95% CI [0.56–1.14], $P = 0.22$) (Fig. S5B).

(3) Further subgroup analyses showed (*Rischin et al., 2021*; *Gillison et al., 2019*; *Gebre-Medhin et al., 2021*) that the incidence of grade III and IV adverse reactions, such as nausea ($P < 0.0001$), vomiting ($P = 0.02$), dehydration ($P < 0.00001$), and fatigue ($P = 0.06$), was significantly higher in the cisplatin group than in the cetuximab group. However, grade III and IV incidence of oral mucositis ($P = 0.02$) and acne-like rash ($P < 0.00001$) was significantly higher in the cetuximab group than in the cisplatin group. The grade III and IV incidence of dysphagia ($P = 0.12$), radiation dermatitis ($P = 0.08$), and hearing impairment ($P = 0.1$) were not significantly different between the two groups (Fig. 8).

## Publication bias detection and sensitivity analysis

Because of the limited number of literature included in the outcome indicators analyzed in this study ($n < 10$), publication bias was not evaluated by the Egger method. Sensitivity analysis of outcome indicators with obvious heterogeneity found that excluding individual studies one by one did not affect the results, suggesting that the conclusions of each outcome indicator were stable and reliable.

## DISCUSSION

The study showed that patients with HPV$^+$ OPSCC in the cetuximab + radiotherapy group had significantly shorter OS and PFS compared to those in the conventional cisplatin + radiotherapy group ($P < 0.05$). Additionally, the incidence of LRF and DM was significantly increased ($P < 0.05$), but there was no significant difference in the overall incidence of grade 3–4 AEs.

This systematic evaluation and Meta-analysis included 5 analyses of the efficacy and safety of cetuximab + radiotherapy regimens compared to cisplatin + radiotherapy for HPV$^+$ OPSCC. De-escalation is a strategy proposed to improve patient's quality of life without significantly compromising the overall treatment efficacy. This study investigated the use of cetuximab + radiotherapy *vs* cisplatin + radiotherapy for patients with HPV$^+$ OPSCC. The meta-analysis results showed that the OS of patients in the cetuximab + radiotherapy group was significantly lower than that in the cisplatin + radiotherapy group. The lower OS may be attributed to the shorter PFS and higher incidence of LRF and DM in the secondary outcome indicators. There was no significant difference in the

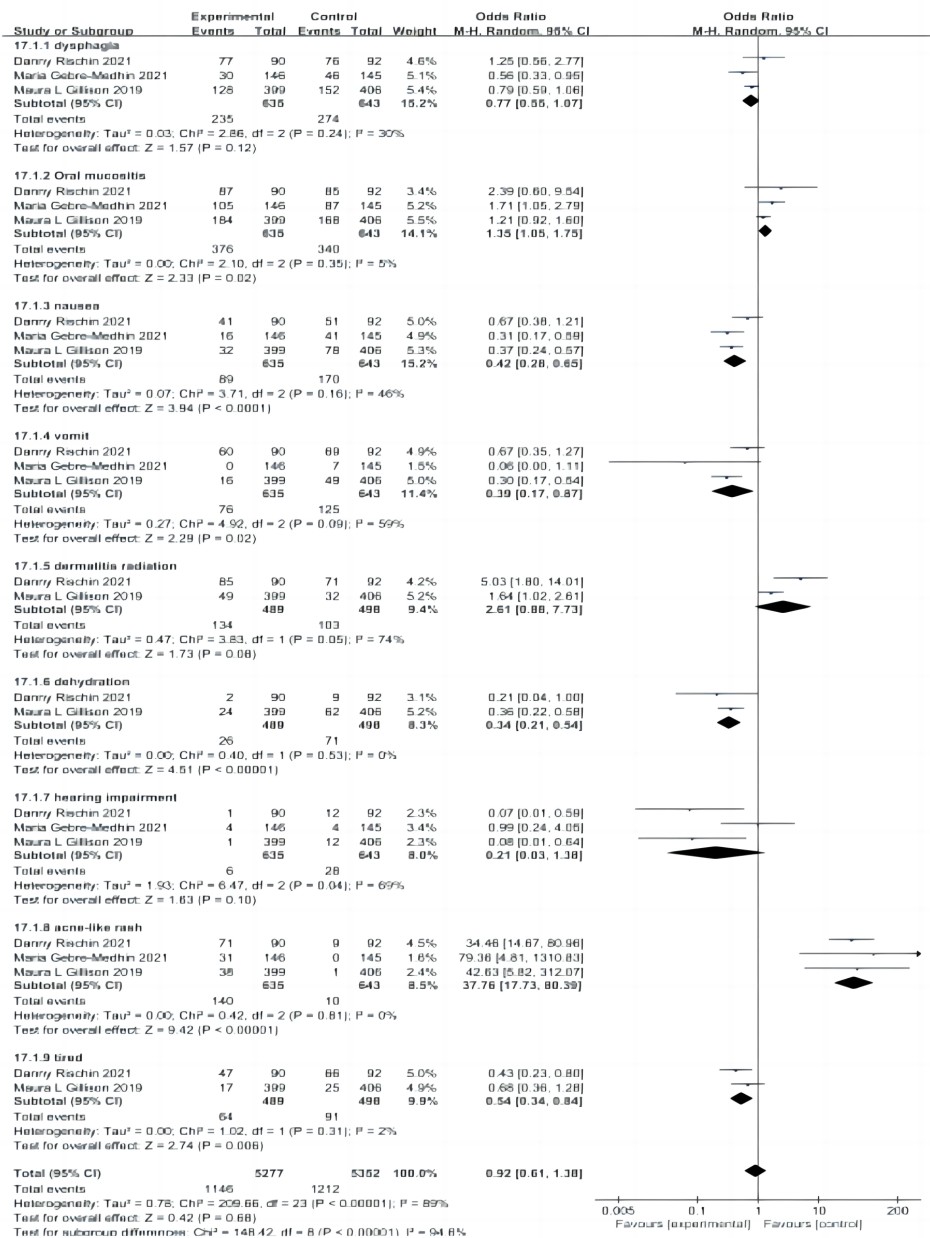

**Figure 8 Forest plot of grade III and IV adverse effects.**

overall incidence of AEs between the groups. However, the group receiving cetuximab + radiotherapy had a significantly higher incidence of grade III and IV AEs, including oral mucositis and acne-like rash.

Is it necessary to completely reject the use of cetuximab instead of cisplatin combined with radiotherapy? Although the analysis above showed a significant trend of impairment in the OS of HPV$^+$ patients for at least 2 years, subgroup analysis revealed no significant

difference in OS between the use of cetuximab + radiotherapy and cisplatin + radiotherapy in early stage non-pharyngeal tumors in patients with no history of smoking or ≤10 packs/year of smoking under the criteria of the 7th edition of the UICC/AJCC and without lymphatic metastases involved. This could be an alternative downstaging approach for patients who fit this profile and are intolerant to cisplatin, particularly those who are allergic to platinum-based agents, following carboplatin/5-fluorouracil treatment. It requires complex continuous indwelling intravenous administration and is associated with more myelotoxicity and mucositis. As a result, it is classified as a recommended regimen in the 2021 edition of the NCCN with a class 2B rating (*National Comprehensive Cancer Network, 2021*).

Limitations of this study: (1) The study group had a small number of cases, and due to the update of the UICC/AJCC version, there was a bias in tumor staging in some patients. Additionally, there may have been a selection bias, which led to the conclusion of some risk factors, such as smoking history, that contradicted previous knowledge. In the discussion of patients with OS, we grouped them according to the classification based on the first infusion of cisplatin and sought the source of the high heterogeneity in the group receiving 100 mg/m$^2$ cisplatin for the first time. Given the difference in the dose of cisplatin in the control group in different studies, we classified it as "Deviations from intended interventions" in the risk quality of bias assessment. Although an "unclear risk of bias" was found in the results of a single risk assessment, it was given a "low risk of bias" in the overall risk assessment. Similarly, the follow-up time of the included studies ranged from 2 to 5 years, but all met our inclusion criteria of more than 2 years. This difference was classified as "selection of the reported outcome" and a "low risk of bias" was obtained in the individual and overall risk assessment. It can be seen that the risk posed by the above differences in this study is acceptable; (2) Some literature may be biased in identifying patients with HPV$^+$ OPSCC. The latest guidelines suggest that using immunohistochemistry alone to define HPV$^+$ OPSCC by P16 expression may lead to false-positive results. Additionally, to ensure accurate results, it is recommended to use RNA scope E6/E7 mRNA *in situ* hybridization as the gold standard for HPV detection and to confirm transcriptionally active HPV in all evaluable cases. This will help to exclude false-positive patients from affecting the experimental results (*Young et al., 2020*). It should be noted that publication bias detection was not performed due to the limited number of literature included in the study; (3) The criteria for classifying HPV$^+$ OPSCC were updated in the 8th edition of the UICC/AJCC staging. However, the treatment regimen should still be validated based on the 7th edition of the staging. The new staging criteria mainly provide a classification of patient prognosis. Until clinical trials validate alternative treatments based on the 8th edition staging, they cannot be used to guide patient management at this time. (4) Our analysis revealed a significant gender disparity in patients with HPV$^+$ OPSCC, with men being diagnosed five times more frequently than women (*D'Souza, McNeel & Fakhry, 2017*). Gender differences have been observed in many tumorigenesis and immunotherapy studies (*Di Donato et al., 2021*). However, none of the experiments mentioned above analyzed the influence of gender on the results, and it remains to be seen whether different genders have significant differences in the prognosis of down-staging treatment for HPV.

We anticipate that additional high-quality randomized controlled clinical trials will be conducted in the future to address the limitations of this study and to provide more robust evidence regarding the efficacy and safety of cetuximab + radiotherapy for the treatment of HPV$^+$ OPSCC. In summary, cisplatin + radiotherapy remains the gold standard for HPV$^+$ OPSCC. Whether cetuximab + radiotherapy can be used instead of cisplatin combination radiotherapy for downstaging patients with localized tumors in early stages who meet specific characteristics intolerant to cisplatin, especially allergic to platinum-based drugs, needs more prospective clinical trials to verify.

### Funding
The authors received no funding for this work.

### Competing Interests
The authors declare there are no competing interests.

### Author Contributions
- Qiong Hu conceived and designed the experiments, performed the experiments, analyzed the data, prepared figures and/or tables, authored or reviewed drafts of the article, and approved the final draft.
- Feng Li conceived and designed the experiments, performed the experiments, analyzed the data, authored or reviewed drafts of the article, and approved the final draft.
- Kai Yang conceived and designed the experiments, authored or reviewed drafts of the article, and approved the final draft.

### Data Availability
This is a systematic review/meta-analysis.

### Supplemental Information
Supplemental information for this article can be found online at http://dx.doi.org/10.7717/peerj.17391#supplemental-information.

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
