# Peer review of "Systematic evaluation and meta-analysis of the prognosis of down-staging human papillomavirus (HPV) positive oropharyngeal squamous cell carcinoma using cetuximab combined with radiotherapy instead of cisplatin combined with radiotherapy"

_PeerJ, doi:10.7717/peerj.17391_

## Round 0.1 · original submission · Minor Revisions

Dear authors, many thanks for your submission. It is unanimous that your work offers relevant information for both clinical decision-making and future research perspectives. However, in order to improve clarity and overall quality, please review the reviewers' reports and respond accordingly.

Reviewer 1 ·

Basic reporting

The authors provided an interesting meta analysis of HPV+ OPC treated with RT + CTX instead of CDDP + RT.
The article is well written, easy to understand and unambiguous. Literature references are adequate in my opinion. Conclusions are ok with the hypotheses.

Experimental design

No comment

Validity of the findings

No comment

Reviewer 2 ·

Basic reporting

This manuscript affirms that cisplatin combination radiotherapy remains the standard of care for HPV+ oropharyngeal squamous cell carcinoma (OPSCC) patients while cetuximab combination radiotherapy did not show any advantages by performing a literature search and analysis. Yet, to ensure our future readers fully appreciate the core findings in this study, some additions and elaborations are highly recommended.

Experimental design

(1) The inclusion criteria seem extremely narrow with a very specific cetuximab regimen and a particular edition of the CTCAE to categorize adverse events. Were all of those inclusion criteria necessary? Further, along this line, as indicated in line 254, a single-arm trial without a control group was eliminated. Is there any concern there? It seems now more and more clinical trials do not require a pure control group since it may not be ethical to do so. Also, the inclusion criteria described in lines 202-204 seem contradictory to the results in lines 273-274.
(2) Further, the majority part of the inclusion and exclusion criteria provided in Section 2.2 seem aimed at enrolling patients instead of screening for published literature suitable for the meta-analysis as proposed. Also, is the sixth one for exclusion accurate?
(3) Also, it seems studies other than the selected reference 20-24 also discussed the topic here, see, for example, reference 17 as discussed in lines 141-148. Would the authors mind clarifying why it was not included in the current meta-analysis? Further along this line, lines 148 to 151 mentioned an earlier meta-analysis, would the author mind confirming all original studies revealed there are also included in the current study? If not, kindly elaborate the reasons behind this exclusion.
(4) Since only the average age was in the inclusion criteria (line 227), would the authors mind clarifying how many of the five studies provided data from age subgroups? If not all five, how many patients were involved in the subgroup analysis shown in the paragraph starting in line 227? This comment generally applies to other subgroup analyses in Section 3.3.1. It would be helpful to know what kind of subgroup information was provided in the five studies and if there is any difference among the studies.

Validity of the findings

(5) Would the author mind elaborating on the sentence starting in line 299? Does it mean no matter what the ECOG score was (and without considering the combination regimen given the sentence immediately following it), the patients’ OS showed no significant difference? If so, is it a little surprising, since I would assume a worse ECOG score showed the patient was in really bad shape and a not good starting point for any regimen?
(6) Further to (5) detailed above, would the author mind clarifying “the results of the above factors combined separately”? Does it mean combining ECOG with smoking history, or ECOG with primary tumor site, or smoking history with primary tumor site? Also, a quick summary of actual patient numbers of those having an ECOG score of 0 or above 0 would be appreciated.

Additional comments

(7) It is suggested to use the past tense in line 166.
(8) According to Figure 1, (805+39=)844 records were returned after the searching, while more records (i.e., 866) were arrived at after removing the duplicates. Similarly, 45 full-text articles were excluded, yet only (12+14+13+1=) 40 of them were given a general reason in Figure 1. Further after excluding 45 articles from 51, 5 were in the final analysis, while 1 was missing. Would the authors mind clarifying the issues here?

Reviewer 3 ·

Basic reporting

The manuscript is well-structured with a comprehensive introduction that contextualizes the research within the existing literature. There are a few grammatical errors, and the manuscript requires a thorough revision.

The methods, results, and discussion sections are logically organized, and the figures and tables are relevant and well-presented. The literature is well-referenced and relevant, and the study complies with PeerJ's requirement for raw data availability.

Experimental design

Transparency in Methodology: Clarify any methodological choices, such as the criteria for study selection and the statistical methods used for pooling data, to enhance the reproducibility of the meta-analysis.

Limited Data on Specific Subgroups: If the original studies lack detailed subgroup analyses, the meta-analysis does not fully capture the nuances of treatment effects across different patient demographics or tumor characteristics.

Validity of the findings

Broader Implications for Clinical Practice: Provide a more detailed analysis of how the findings can be integrated into current clinical guidelines for treating HPV+ oropharyngeal squamous cell carcinoma, considering different healthcare settings.

Subgroup Analysis: If data permits, include subgroup analysis to identify specific patient populations that may benefit more from cetuximab combination therapy compared to cisplatin combination therapy.

Consideration of Recent Literature: Update the literature review to include the most recent studies that might have been published after the initial search was conducted, ensuring that the meta-analysis reflects the current state of research.

Additional comments

Enhanced Discussion on Limitations: Expand the section discussing the limitations of the study, including study heterogeneity, potential biases in selected studies, and the impact of these factors on the results and conclusions.

---

## Round 0.2 · Minor Revisions

Dear authors, many thanks for your resubmission and hardwork. Your manuscript still requires minor revisions in order to be ok for publication. Please, refer to the reviewer's comments for further details. I'd also like to remind you to fully proofread the whole document and materials and respect the journals requirements for figures quality (in case you haven't done so already).

Reviewer 2 ·

Basic reporting

Many thanks for the authors’ patient elaboration and amendments to the manuscript. Really appreciate them. Yet, I do have some minor concerns as detailed below.

Experimental design

(1) To avoid potential confusion, it is highly suggested to at least modify the sentences in Section 2 to clarify that the criteria were set to include or exclude studies (or studies investigating certain patients with certain treatments) instead of to include or exclude patients themselves. For example, the first exclusion criteria state that “[t]hose who had not been diagnosed by histopathology,” while the third says “[r]eviews, systematic evaluations, meta-analyses, letters, and case reports.”

(2) Further considering the pretty narrow inclusion criteria, it is highly suggested to at least remind our readers in some sections outside of Section 2. For example, in the “Eligibility criteria for selecting studies” section of the abstract, is it possible to state “[w]e included randomized controlled trials reporting results of standard regimens of cetuximab + radiotherapy vs cisplatin + radiotherapy in treating HPV+ OPSCC” or similar?

(3) Again, Section 3.3.1 compares the OS outcome between those having the first dose of cisplatin dose of 70 mg/m2 vs 100 mg/m2. Yet, the 4th inclusion criteria states the cisplatin dose should be 250 mg/m2. The authors’ answer seems to focus on cetuximab instead of cisplatin. Even considering cetuximab may be administrated at the first dose ranging from 70-100 mg/m2, this range is not clarified in the 3rd inclusion criteria. Instead, it is only said “400 mg/m2 cetuximab intravenously 5-7 days prior to the start of radiotherapy, followed by 250 mg/m2 cetuximab intravenously 7 times per week during the radiotherapy period (total 2,150 mg/m2)”. Since this study does not meet the 3rd and the 4th inclusion criteria, shall it be excluded from the current analysis? If those are just certain “exception” subgroups of the study while the study does contain other subgroups that meet the inclusion criteria, it would be highly recommended to emphasize it in the manuscript. Otherwise, it would be really confusing that those “exception” subgroups constitute a significant part of the result while other studies focusing on similar “exceptions” are excluded from the current analysis. After reading the whole manuscript, I have to say that my best guess is that the current inclusion criteria #3 and #4 are not described accurately. Authors' clarification here and further amendments to the manuscript would be really appreciated.

Validity of the findings

NA

Additional comments

(4) It is suggested to rewrite the last sentence of the conclusion section in the abstract since it is a little confusing.

(5) In the second paragraph of the background section, the following sentence fragment was repeated twice: “generally have a better prognosis compared to patients with HPV- OPSCC.”

(6) In the third paragraph of the background section, the “《” and “》” marks are not generally used in English, I believe. Also, it seems that the “,” immediately after these marks and the reference to [9] should be “.” instead. Further, for ease of reading, it is suggested to separate the third sentence in this paragraph into multiple ones.

(7) Does it make sense to change “XXX combination radiotherapy” to “XXX combined with radiotherapy” or simply “XXX + radiotherapy” across the manuscript?

(8) The third sentence of Section 2.3, the first sentence of the second paragraph in Section 3.3.1, and the first sentence of the third paragraph in Section 3.3.1 seem missing a verb.

(9) The sentence in the discussion section stating “[d]e-escalation is a strategy proposed to reduce patients' OS without compromising their treatment” (emphasis added) is really confusing since it is very hard for me to believe that we need a strategy to reduce patients’ overall survival. Are the authors meant to say that “de-escalation is a strategy proposed to improve patient’s quality of life without significantly compromising the overall treatment efficacy” or similar?

---

## Round 0.3 · accepted · Accept

Dear authors, many thanks for your careful responses and diligent manuscript improvement. I am happy to let you know that i am now accepting it for publication in PeerJ. best of success.